# Therapeutic Vulnerability to ATR Inhibition in Concurrent *NF1* and *ATRX*-Deficient/ALT-Positive High-Grade Solid Tumors

**DOI:** 10.3390/cancers14123015

**Published:** 2022-06-19

**Authors:** Ming Yuan, Charles G. Eberhart, Christine A. Pratilas, Jaishri O. Blakeley, Christine Davis, Marija Stojanova, Karlyne Reilly, Alan K. Meeker, Christopher M. Heaphy, Fausto J. Rodriguez

**Affiliations:** 1Department of Pathology, Johns Hopkins University School of Medicine, Baltimore, MD 21218, USA; myuan3@jhmi.edu (M.Y.); ceberha@jhmi.edu (C.G.E.); cmd337@gmail.com (C.D.); alan.meeker@gmail.com (A.K.M.); 2Department of Oncology, Johns Hopkins University School of Medicine, Baltimore, MD 21218, USA; cpratil1@jhmi.edu (C.A.P.); jblakel3@jhmi.edu (J.O.B.); 3Department of Pediatrics, Johns Hopkins University School of Medicine, Baltimore, MD 21218, USA; 4Department of Neurology, Johns Hopkins University School of Medicine, Baltimore, MD 21218, USA; 5Department of Medicine, Boston University School of Medicine and Boston Medical Center, Boston, MA 02118, USA; marija.stojanova@bmc.org; 6National Cancer Institute, Bethesda, MD 21218, USA; reillyk@mail.nih.gov; 7Department of Pathology and Laboratory Medicine, Boston University School of Medicine and Boston Medical Center, Boston, MA 02118, USA; 8Department of Pathology and Laboratory Medicine, David Geffen School of Medicine, University of California Los Angeles (UCLA), 10833 Le Conte Avenue, CHS Bldg., Suite 18-170B, Los Angeles, CA 90095, USA

**Keywords:** ATRX, alternative lengthening of telomeres, neurofibromatosis

## Abstract

**Simple Summary:**

Tumors of the brain and nerves develop frequently in patients with neurofibromatosis type 1. Many are benign growths, such as pilocytic astrocytomas in the brain and neurofibromas in the nerves. However, in some patients, the tumors become malignant and may cause local damage, disseminate to distant sites and result in death. We studied changes in the levels of chromatin proteins and changes in telomeres, in cells obtained from mouse gliomas that are deficient in neurofibromin as well as excess brain and nerve tumor tissue from patients with neurofibromatosis type 1 or sporadic tumors lacking neurofibromin expression. A decrease in the levels of these proteins in experimental cell lines resulted in susceptibility to a class of specific drugs knowns as ATR inhibitors, which may represent a specific vulnerability of these tumor subgroups. We expect our data to provide the required rationale for the development of more accurate animal models to study neurofibromatosis, as well as specific molecularly based drugs for treatment as alternatives to the current, often devastating approaches of surgery, radiation, and chemotherapy.

**Abstract:**

Subsets of Neurofibromatosis Type 1 (NF1)-associated solid tumors have been shown to display high frequencies of ATRX mutations and the presence of alternative lengthening of telomeres (ALT). We studied the phenotype of combined NF1 and ATRX deficiency in malignant solid tumors. Cell lines derived from NF1-deficient sporadic glioblastomas (U251, SF188), an NF1-associated ATRX mutant glioblastoma cell line (JHH-NF1-GBM1), an NF1-derived sarcoma cell line (JHH-CRC65), and two NF1-deficient MPNST cell lines (ST88-14, NF90.8) were utilized. Cancer cells were treated with ATR inhibitors, with or without a MEK inhibitor or temozolomide. In contrast to the glioma cell line SF188, combined ATRX knockout (KO) and TERC KO led to ALT-like properties and sensitized U251 glioma cells to ATR inhibition in vitro and in vivo. In addition, ATR inhibitors sensitized U251 cells to temozolomide, but not MEK inhibition, irrespective of ATRX level manipulation; whereas, the JHH-NF1-GBM1 cell line demonstrated sensitivity to ATR inhibition, but not temozolomide. Similar effects were noted using the MPNST cell line NF90.8 after combined ATRX knockdown and TERC KO; however, not in ST88-14. Taken together, our study supports the feasibility of targeting the ATR pathway in subsets of NF1-deficient and associated tumors.

## 1. Introduction

Neurofibromatosis type 1 (NF1) is an inherited tumor predisposition syndrome, and individuals affected with NF1 are prone to develop tumors of the central nervous system (CNS) and peripheral nervous system. Rarely, these individuals may develop other solid neuroendocrine and mesenchymal tumors, some of which may behave in a malignant fashion [1]. The most common primary CNS tumors in these patients are pilocytic astrocytomas (PA), which have a predilection for the optic pathways, especially in children. It is also known that the full spectrum of glial neoplasia may develop in these patients, including low- and high-grade astrocytomas [2]. Specific drivers of sporadic high-grade astrocytomas have been elucidated in the past years through genome-wide sequencing [3], and similar techniques have been applied to study NF1-associated astrocytomas, identifying genetic alterations in *TP53, CDKN2A,* and *ATRX* [4].

Telomeres are composed of a repetitive DNA sequence (TTAGGG) bound by a shelterin protein complex that protects the ends of linear chromosomes to maintain genomic integrity. However, in normal cells, telomeres progressively shorten with each cell division. Critically short telomeres can result in breakage–fusion–bridge cycles that may lead to the accumulation of catastrophic global genomic damage or cellular senescence. To ensure unlimited replication potential, cancer cells employ two major, largely mutually exclusive, mechanisms of telomere maintenance. The predominant mechanism is expression of the reverse transcriptase, telomerase, which is encoded by an RNA component (*TERC*) and the rate-limiting catalytic subunit (*TERT*). In cancer, telomerase up-regulation is frequently mediated through activating mutations [5], amplifications, structural variants, and promoter methylation [6] in *TERT*. However, a subset of cancers maintains its telomere lengths independent of telomerase, using the *Alternative Lengthening of Telomeres* (ALT) [7]. ALT results from an aberrant homologous recombination-based process mediated by homology-directed repair that leads to the presence of unique molecular features in ALT-positive cancers, including the presence of ultra-long telomeres, dramatic cell-to-cell telomere heterogeneity (assayable in cell and tissue samples via telomere-specific FISH), and the presence of single-stranded extrachromosomal circles containing the C-rich telomere repeat sequence (C-circles; assayable by rolling circle amplification). Variations in telomere length have emerged as a prognostic factor in a variety of tumor types [8,9]. Our group has reported a high frequency of ALT in high-grade astrocytomas developing in individuals with NF1-syndrome, as well as a small subset of MPNST, although usually not in their benign counterparts [10].

Prior studies have linked alterations in the *alpha thalassemia/mental retardation syndrome X-linked (ATRX)* or *death domain-associated protein (DAXX)* genes with ALT in a subset of cancers [11]. Loss of ATRX function leads to abnormal methylation and gene expression patterns, as well as chromosome mis-segregation. In the nucleus, ATRX cooperates with the molecular chaperone DAXX to incorporate the H3.3 histone variant in heterochromatic regions, including at telomeres. *ATRX* mutations and ALT are associated with specific molecular subgroups of sporadic brain tumors [4,10]. In NF1-associated tumors, particularly astrocytomas, *ATRX* mutations and ALT are associated with specific tumor groups such as the recently recognized WHO tumor type high-grade astrocytoma with piloid features [12,13].

Prior studies have documented that ATRX loss and ALT may represent a therapeutic vulnerability [14,15] and that these cells are sensitive to ATR inhibition in sporadic tumorigenesis [16]. Additionally, ATRX inactivation promotes DNA damage and cellular death, which may synergize with specific therapeutic approaches. Thus, we hypothesized that the *ATRX* mutations and telomere alterations that occur in distinctive subsets of NF1-associated tumors, particularly those with an aggressive phenotype, represent a potential vulnerability that can be therapeutically targeted.

## 2. Materials and Methods

### 2.1. Cell Lines

Human tumor specimens were collected at Johns Hopkins Hospital with local Institutional Review Board approval, and written informed consent was obtained from patients or their parents. Tissues were minced and digested with either papain dissociation system Grand Island, New York, NY, USA), then filtered through a 70 μm Falcon cell strainer (ThermoFisher Scientific, Waltham, MA, USA). JHH-NF1-GBM1 was grown in DMEM/F12 containing 1× B27 supplement (ThermoFisher Scientific), 20 ng/mL EGF (PeproTech, Cranbury, NJ, USA), 20 ng/mL FGF-b (PeproTech), and 5 µg/mL Heparin (Millipore SIGMA, Burlington, MA, USA). Conditionally reprogramming culture (CRC) cell lines (JHH-NF1-PA1 and JHH-CRC65) was grown under the conditions described previously [17]; 50% F medium (25% F-12, 75% DMEM supplied with 10% FBS and 5 µg/mL Insulin), 50% 3T3 condition medium supplied with 25 ng/mL Hydrocortisone (Millipore Sigma), 0.1 nmol/l Cholera toxin (Millipore SIGMA), 10 ng/mL EGF (PeproTech), and 5 µM ROCK inhibitor Y-27632 (Selleckchem, Houston, TX, USA). U251 was purchased from the American Type Culture Collection (ATCC) and originally derived from an adult glioblastoma patient and characterized by typical adult GBM alterations including gains of chromosomes 3, 7, 15, and 17 and losses of chromosomes 10, 13, and 14, as well as an *NF1* nonsense mutation (NF1 c.2033dupC) [18]. SF188 was kindly provided by Dr. Chris Jones (Institute of Cancer Research, Sutton, UK) [19] and is derived from a pediatric high-grade glioma. At the genetic level, it contains amplifications of *MYC* at 8q24, *CCND1* at 11q13, and *CDK4* at 12q14, which are frequently found in pediatric high-grade gliomas. It also has a deletion of *NF1* at 17q11.2. U251 ATRX^−/−M^, U251 ATRX^−/−2.02^, SF188 ATRX^−/−F^, and SF188 ATRX^−/−N^ were utilized as previously described [20]. U251 and SF188 cells were grown in DMEM/F12 supplemented with 10% FBS (ThermoFisher Scientific). MPNST-derived cell lines NF90-8 and ST88-14 were maintained in RPMI supplemented with 10% FBS (ThermoFisher Scientific). Murine glioma cell lines 130G#3, 158D#8, 1491-9, and 1861-10 were maintained in DMEM supplemented with 10% FBS (ThermoFisher Scientific) [21]. All cells were cultured in a humidified 37 °C incubator with 5% CO_2_. Human Schwann cells were purchased from ScienCell and maintained in Schwann cell media. Cell lines were routinely tested for mycoplasma and human cell line identities were confirmed by short tandem repeat (STR) profiling (Johns Hopkins University Genetic Resource Core Facility, Baltimore, MD, USA).

### 2.2. Quantitative Real-Time Polymerase Chain Reaction (qRT-PCR)

Total RNA was isolated from cultured cells using the RNeasy mini kit (QIAGEN), and cDNAs were produced using QuantiTect reverse transcription kit (QIAGEN). qRT-PCR was performed using PowerUp SYBR Green Master Mix (ThermoFisher Scientific). Primer sequences were ATRX: forward 5′- CAATCACAGAAGCCGACAAG -3′, reverse 5′- GTCATGAAGCTTCTGCACCA -3′; BCL2 forward 5′- GGACAAGTGCAGGAGTGGAT -3′, reverse 5′- CGTCCCCGTATAGAGCTGTG -3′; CDKN1A forward 5′- AGTCAGTTCCTTGTGGAGCC -3′, reverse 5′- CATGGGTTCTGACGGACAT -3′; CDKN1B forward 5′- AAGAAGCCTGGCCTCAGAAG -3′, reverse 5′- TTCATCAAGCAGTGATGTATCTGA -3′; CDKN2A forward 5′- GTTACGGTCGGAGGCCG -3′, reverse 5′- GTGAGAGTGGCGGGGTC -3′; MCL1 forward 5′- AGACCTTACGACGGGTTGG -3′, reverse 5′- TCCTGATGCCACCTTCTAGG -3′; PARP1 forward 5′- GATGGGTTCTCTGAGCTTCG -3′, reverse 5′- TCTGCCTTGCTACCAATTCC -3′; TERC forward 5′- CCCATTCATTTTGGCCGACTT -3′, reverse 5′- GGCCGCTCCCTTTATAAGC -3′; HPRT1 forward 5′- GTTATGGCGACCCGCAG -3′, reverse 5′-ACCCTTTCCAAATCCTCAGC -3′. HPRT1 was used as the endogenous control. The relative fold changing was calculated based on the formula R = 2^−(ΔCt sample−ΔCt control)^. Quantification qPCR assay for telomerase activity was performed according to the manufacturer’s instructions (#8928, ScienCell).

### 2.3. Gene Knockdown and Knockout of ATRX or TERC

The CRISPR cas9 nickase system was used to generate inactivating mutations in either the *ATRX* or *TERC* genes, as previously reported [22,23]. Guide RNAs (gRNAs) targeting TERC were obtained from Abmgood. Short hairpin RNA target human ATRX (sh11 and sh90), mouse ATRX (sh1 and sh3), and vector control pLKO.1 were obtained from Millipore Sigma. To produce lentiviruses, 293T cells were transfected with shRNA or gRNA plasmid and VSVG packaging plasmids mixture using lipofectamine 2000 (ThermoFisher Scientific). Lentiviral supernatants were collected 48–72 h later and kept frozen at −80 °C until needed. Cells infected with the virus were selected with 1–2 μg/mL of puromycin (MilliporeSigma), 2–4 μg/mL of Blastidin (ThermoFisher Scientific), and 100–400 μg/mL of G418 (ThermoFisher Scientific) for 7 days to generate stable cell lines. ATRX knockdown was confirmed by Western blotting and *TERC* knockout (KO) was confirmed by qPCR.

### 2.4. ALT Validation

ALT was assessed with previously established methods [24,25,26]. Ultra-bright telomeric foci were evaluated by telomere-specific FISH. Telomeric extra-chromosomal circles (e.g., C-circles) were detected using immunoblotting after a processive phi29 polymerase to amplify C-circle DNA. A DIG-conjugated probe containing the C-rich telomere repeat sequence specifically targeted the polymerase amplified signal. The known ALT-positive osteosarcoma cell line U2OS served as a positive control.

### 2.5. Cell Growth Assessment

To assess effects on cell growth, the CellTiter-Blue assay was used (Promega). In brief, 1000 to 5000 cells were plated in triplicate in 96-well plates. Then, 20–30 μL of the CellTiter-Blue reagent was added per well in 96-well plates and incubated for 1–4 h at 37 °C in 5% CO_2_. For drug treatments, cells in 96-well plates were cultured with various drug concentrations of AZD6244, AZD6738, VE-822, or temozolomide (Selleckchem). Vehicle (Dimethyl sulfoxide)-treated cells were used as controls and the cell survival fraction was calculated as a percentage of control cells. Fluorescence (560 nmEx/590 nmEm) was measured using a TECAN plate reader. Additionally, apoptosis assays were performed using Muse Annexin V and Dead Cell reagent (MilliporeSigma), and Bromodeoxyuridine (BrdU) incorporation assays were performed as previously described [27]. Data were acquired using Muse flow cytometer (Millipore) and analyzed with FlowJo software.

### 2.6. Western Blotting

Cells were lysed in RIPA lysis buffer supplemented with protease inhibitors (MilliporeSigma). Primary antibodies used for Western blots were: NF1 (A300-140A, 1:1000, Bethyl Laboratories, Montgomery, TX, USA), ATRX (#10321, 1:500, Cell Signaling Technology, Danvers, MA, USA), α-tubulin (#3873, 1:5000, Cell Signaling Technology), β-actin (sc-47778, 1:5000, Santa Cruz Biotechnology, Dallas, TX, USA), pERK1/2 (#4370, 1:1000, Cell Signaling Technology), Erk1/2 (#9102, 1:1000, Cell Signaling Technology), and γH2AX (#9718, 1:1000, Cell Signaling Technology). Secondary antibodies used for Western blots were anti-mouse IgG HRP-linked (#7076, 1:5000, Cell Signaling Technology) and anti-rabbit IgG HRP-linked (#7074, 1:5000, Cell Signaling Technology). Original WB images can be found at Appendix A.

### 2.7. MPNST Xenograft

For in vivo experiments, 1 × 10^6^ cells were sciatic nerve transplanted in Nude mice (Charles River, Wilmington, MA, USA). For bioluminescence imaging, cells were labeled with the lentiviral-based reporter co-expressing RFP and luciferase (SBI). Animals were closely monitored for tumor growth, and euthanized when tumor size was over 1000 mm^3^.

### 2.8. In Vivo Drug Testing

For in vivo experiments, 5 × 10^5^ glioma cells were orthotopically transplanted (Stereotaxic coordinate: X(AP) = 1.0 mm, Y(ML) = 2.0 mm, Z(DV) = −3 mm) in Nude mice (Charles River). For bioluminescence imaging, cells were labeled with the lentiviral-based reporter co-expressing RFP and luciferase (SBI). Oral drugs were delivered to xenografted mice once daily by gavage from d2 to d16. Animals were closely monitored for tumor growth, and euthanized when neurologic signs of disease develop. In addition, subsets were sacrificed at appropriate intervals and tissue sections microscopically examined for early evidence of tumor formation. Methods for the analysis of tumor xenograft for morphology, size, proliferation, and differentiation have previously been reported [28,29,30].

### 2.9. Statistical Analyses

For cell culture and functional assays, data were presented as mean ± standard deviation with *p* < 0.05 considered statistically significant. All experiments were performed in at least three biological replicates and data analyzed with a two-tailed Student’s t-test or ANOVA as appropriate. Survival was assessed with Kaplan–Meier curves and statistical analyses made using standard software and statistical packages (GraphPad, San Diego, CA, USA).

## 3. Results

### 3.1. ATRX Loss in the Context of Diminished Telomerase Activity Facilitates the Development of ALT-Associated Hallmarks

To identify the biologic relevance of ATRX loss in NF1-associated gliomagenesis, we studied four previously characterized *Nf1*^+/−^*Trp53*^+/−^ murine glioma lines (130G#3, 158D#8, 1491-9, and 1861-10) representing all of the histologic diffuse glioma grades, 2–4 [21]. Compared to NIH-3T3 cells, these cell lines all underexpress Nf1 while demonstrating variable Atrx expression, with Atrx loss in two cell lines (cell lines 1491-9 and 1861-10), and preserved Atrx expression in two cell lines (cell lines 130G#3 and 158D#8) (Appendix A). No significant effects on cell growth were detected after ATRX knockdown in cell lines 130G#3 and 158D#8 (Figure 1 and Appendix A).

Since ALT can develop in the presence of diminished telomerase activity [31], we tested the effects of Atrx knockdown in the 130G#3 cell line in the presence of the telomerase inhibitor, BIBR1532. This inhibitor decreases telomerase activity by binding to the active site of TERT, thereby downregulating TERT expression [32,33]. With the known caveat that murine telomere lengths are significantly longer and display stronger telomeric FISH signals compared to human cancer cells, we identified increased telomere FISH signal brightness, with features reminiscent of ALT, after prolonged Atrx knockdown (55 days) and concurrent Tert inhibition (39 days) (Figure 1).

### 3.2. ATRX Loss Decreases Cell Growth in NF1-Deficient Human Glioma Lines, but with No Effect in MPNST Lines

Next, we proceeded to test the effect of ATRX loss in glioma cells developing in patients with NF1, which are ATRX wildtype and lack ALT. Using the Conditionally Reprogramming Culture (CRC) technique [34], we previously developed a human pilocytic astrocytoma cell line derived from a patient with NF1 [17]. ATRX depletion through shRNA resulted in decreased growth and increased apoptosis (Figure 2). Decreased levels of BCL2, increased levels of MCL1, and increased levels of the senescence markers, CDKN1A and CDKN1B, were also detected (Appendix A). In addition, we detected increased levels of PARP and p-H2AX foci (Figure 2 and Appendix A) suggesting that apoptosis may be the result of replicative stress in these cells, which may have impaired DNA repair mechanisms.

Since ATRX loss and ALT activation are largely limited to high-grade neoplasms in the context of NF1 loss or inherited NF1 syndrome [10], we evaluated cell lines derived from high-grade neoplasms with NF1 loss, including two sporadic glioblastoma cell lines with NF1 inactivation (U251, SF188), an NF1-associated glioblastoma (JHH-NF1-GBM1), two MPNST cell lines (NF90-8, ST88-14), and an NF1-associated sarcoma line (JHH-CRC65). These cell lines displayed NF1 protein loss and preserved DAXX protein expression, but demonstrated variable ATRX protein levels. Notably, ATRX protein expression was higher in the sporadic glioma lines (U251, SF188), while completely absent in the NF1-associated glioblastoma line JHH-NF1-GBM1 (Appendix A).

To assess the effect of ATRX depletion in high-grade glioma cell lines that are NF1 deficient and grow easily, and therefore are more feasible for functional experiments, we selected U251 and SF188 as described above. ATRX KO (via CRISPR) resulted in decreased growth in the high-grade glioma cell lines (U251, SF188), with increased apoptosis more evident in U251 in vitro (Figure 3A–D). Since U251 (but not SF188) develops ALT after ATRX knockdown or KO, we used U251 for subsequent in vivo experiments [20]. Reduced growth of U251 after ATRX KO was also evident in vivo using orthotopic intracranial xenografts upon histological examination (Appendix A), leading to a prolonged survival (Figure 3E). Functional loss of ATRX resulted in ALT in U251, but not in SF188 [20], and ALT features were present in the NF1-glioblastoma line JHH-NF1-GBM1 as demonstrated by the c-circle assay (Appendix A) and telomere-specific FISH.

Next, we studied the effects of ATRX loss in NF1-deficient MPNST and sarcoma cell lines. In contrast to the NF1-deficient glioma lines, and despite successful ATRX knockdown, ATRX loss had no effect on growth in vitro (NF90-8, ST88-14, and JHH-CRC65) or in vivo (NF90-8) (Appendix A). Telomere-specific FISH staining of these cells showed rare ultrabright, ALT-like foci (Figure 4). In contrast, ATRX knockdown resulted in decreased growth of a non-neoplastic Schwann cell line with increased levels of senescence marker p21 (CDKN1A) (Appendix A).

### 3.3. ATRX Deficiency Sensitizes Cells to ATR Inhibitors and Temozolomide

To test the feasibility of ATR inhibition in the treatment of high-grade neoplasms, we evaluated NF1- and ATRX-deficient cancer cells. The glioma cell line U251, which develops ALT after ATRX KO [20], was more sensitive to two different ATR inhibitors (AZD6738 and VE-822) in the absence of ATRX, in contrast to the SF188 cell line, which remains ALT-negative (Figure 5). In addition, ATRX KO also sensitized U251 to MEK inhibition (AZD6244) (Figure 5). MEK inhibition has emerged recently as a targeted therapeutic approach for NF1-associated tumors [35,36], and temozolomide is a standard chemotherapeutic agent for high-grade gliomas [37] and is occasionally used to treat MPNST. Therefore, we next tested the ATR inhibitor, AZD6738, in combination with temozolomide or MEK inhibition. The combination treatment of temozolomide and AZD6738 profoundly decreased growth in ATRX-deficient U251 cells (Appendix A), while MEK inhibition (AZD6244) had no additional effect despite inhibiting MEK signaling as demonstrated by a decrease in pERK levels (Appendix A). The combination treatment of temozolomide and AZD6738 also decreased growth in ATRX-deficient MPNST NF90-8 cells (Appendix A). Next, we proceeded to study a cell line that is NF1 and ATRX deficient and derived from an NF1 patient as a more rigorous model of concurrent NF1 and ATRX deficiency. The NF1-associated glioblastoma cell line JHH-NF1-GBM1, which carries an ATRX mutation and exhibits ALT, grows under neurosphere conditions and intracranially as orthotopic xenografts, albeit slowly (Figure 6A–C). This cell line demonstrates sensitivity to ATR inhibition (AZD6738) in vitro while being resistant to temozolomide, a standard drug used in the treatment of high-grade glioma (Figure 6D,E). Similarly, ATRX knockdown and TERC KO in the MPNST cell line NF90-8 successfully suppressed telomerase activity (Appendix A), and sensitized cells to ATR inhibition (AZD6738 and VE-822) (Figure 7). In contrast, ATRX knockdown in the ALT-negative, NF1-associated sarcoma line JHH-CRC65 did not increase sensitivity to either ATR inhibition or temozolomide treatment (Appendix A).

Finally, we investigated the effect of ATR inhibition and temozolomide treatment on orthotopic glioma xenografts in vivo. Oral administration of the ATR inhibitor, AZD6738, decreased growth transiently in ATRX-deficient U251 xenografts, while the combination of temozolomide plus AZD6738 had a more pronounced and persistent inhibitory effect on tumor cell growth (Figure 8).

## 4. Discussion

Inactivation of ATRX, or, less frequently, its associated protein DAXX, through deleterious mutations, has been increasingly observed in subsets of cancers, and is consistently associated with ALT. Although less prevalent overall in gliomas compared to the mutually exclusive *TERT* promoter mutations, *ATRX* mutations and ALT are frequent in specific molecular subgroups of sporadic brain tumors, particularly IDH mutant astrocytomas [4,10]. In NF1-associated tumors, high-grade astrocytomas in particular have a high prevalence of *ATRX* mutations and ALT, and these alterations are associated with worse outcomes [10]. In this study, ALT and *ATRX* mutations were also present in MPNST, but in a smaller subset and not associated with a difference in outcome when compared to MPNST lacking ALT. In another study of MPNST, ATRX loss detected by immunohistochemistry was associated with worse overall survival; although, the group included partial or “mosaic” loss, which is not always associated with *ATRX* mutations and ALT [38].

There has been increasing interest in the biological effect of ATRX, particularly whether targeted therapies may be helpful for these tumors, which are often high-grade and difficult to treat. Functional loss of ATRX leads to G-quadruplex (G4) DNA secondary structures as a result of replication stress in glioma models [39]. For example, *Atrx*-null neural progenitor cells are exquisitely sensitive to telomestatin, a compound that stabilizes G4 DNA secondary structures [40]. Several groups have also reported on the efficacy of specific drugs for *ATRX*-mutant gliomas. In a mouse model developed using the *Sleeping Beauty* transposon system, Koschmann et al. demonstrated that Atrx deficiency leads to impaired non-homologous end joining, and sensitizes cells to compounds associated with the formation of double-stranded DNA breaks [41]. More recently, this group demonstrated that ATRX-deficient cells are associated with loss of Chk1 and reliant on ATM, thereby suggesting that ATM inhibition may sensitize these cells to radiation therapy [42]. In addition, ATRX depletion in cells results in the persistent association of telomeres with replication protein A, which, in the presence of ATR inhibition, results in the disruption of ALT and selective cell death [16]. Several ATR inhibitors are in active phase I/II [43], including VX-970 and AZD6738, with AZD6738 having the added benefit of being orally bioavailable.

Despite the observation that ATRX-deficient tumors essentially always develop ALT, this is not consistently observed in model systems, suggesting that in certain cellular contexts, ATRX loss is not sufficient to induce ALT [20,23,44]. Our experience demonstrates that attenuated telomerase activity (i.e., telomerase inhibition) is necessary in some models, which may require prolonged or permanent TERT inactivation, and our data suggest this phenomenon to be cell-type dependent. The focus of our study is on the functional loss of ATRX, since it is the predominant gene inactivated by mutations in solid tumors associated with ALT, particularly tumors arising from the central and peripheral nervous systems. It seems that indeed, ALT-positive cells may require the presence of specific DNA repair proteins, but the precise requirements vary by cell type [45].

## 5. Conclusions

ATRX loss and the development of ALT in NF1-deficient tumors represent a potential therapeutic vulnerability to ATR inhibition that may lead to much-needed targeted treatments for solid tumors developing in patients with NF1 or in the sporadic setting. However, this vulnerability may be cancer type and/or cell line dependent, and therefore further studies are needed to identify the best biomarkers and/or drugs that will help manage this patient population. Importantly, these will include other pathways associated with the DNA damage response, such as CHK1, which are increasingly being targeted in oncology.

## Figures and Tables

**Figure 1 cancers-14-03015-f001:**
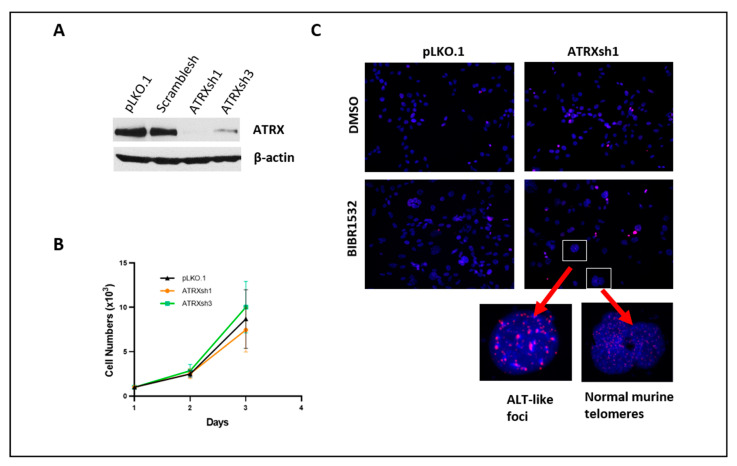
Atrx knockdown in *Nf1*^+/−^*Trp53*^+/−^ murine glioma cell line 130G#3. Reduction in Atrx expression using Atrx shRNAs was performed (**A**) and there were no significant effects on cell growth (**B**). Development of ultrabright telomere foci, reminiscent of ALT, after Atrx knockdown and treatment with the telomerase inhibitor, BIBR1532 for 39 days (**C**). Empty lentiviral vector (pLKO.1) and scramble shRNA were used as controls.

**Figure 2 cancers-14-03015-f002:**
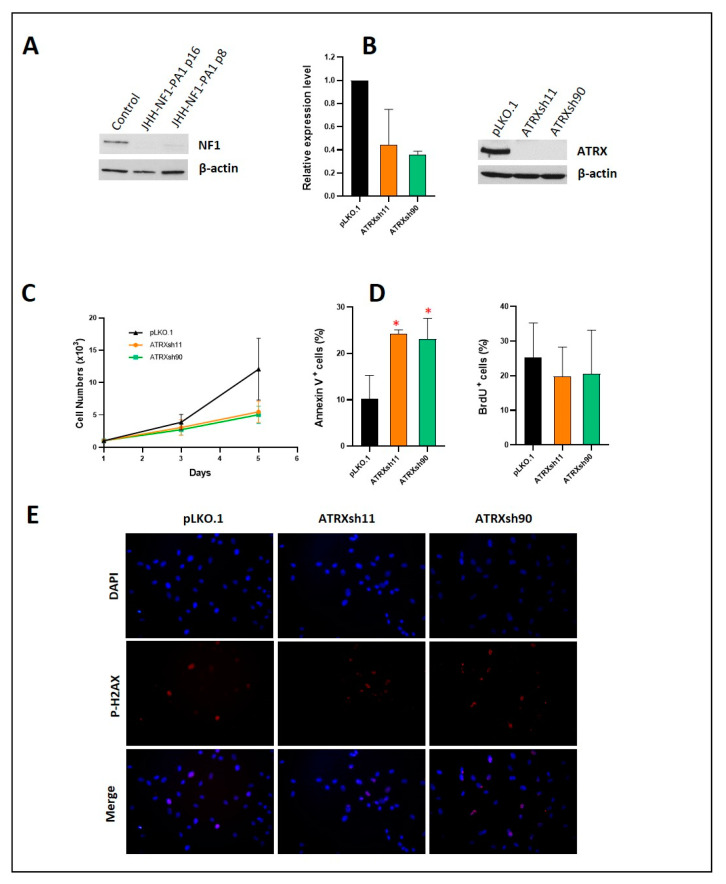
ATRX knockdown in the human glioma cell line JHH-NF1-PA1. NF1 pediatric pilocytic astrocytoma cell line JHH-NF1-PA1 has pronounced NF1 loss in culture (**A**). Successful *ATRX* knockdown (mRNA left, protein right) using shRNA hairpins (**B**). ATRX loss leads to decreased growth of JHH-NF1-PA1 (**C**), primarily through increased apoptosis (**D**). Short-term ATRX loss increased p-H2AX foci (**E**). (* *p* < 0.05).

**Figure 3 cancers-14-03015-f003:**
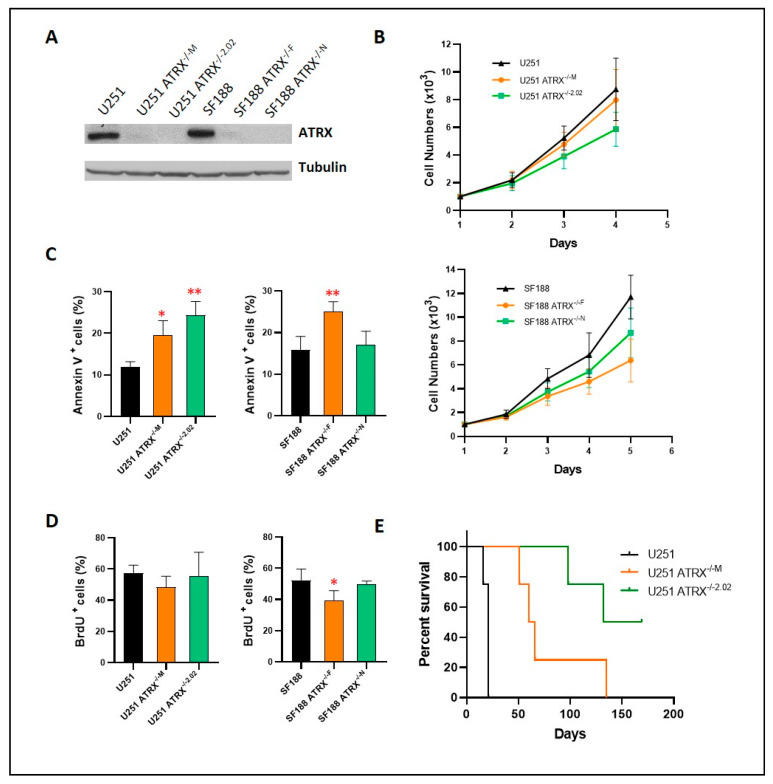
ATRX knockout in NF1-deficient glioblastoma cell lines U251 and SF188: Successful ATRX knockout in both glioma lines at the protein level (**A**). Decreased cell growth, primarily secondary to apoptosis, was noted in vitro (**B**,**C**), with mild to no effect on proliferation (**D**). ATRX knockout in U251 resulted in decreased tumor growth in orthotopic nude mouse xenografts (**E**). (* *p* < 0.05, ** *p* < 0.01).

**Figure 4 cancers-14-03015-f004:**
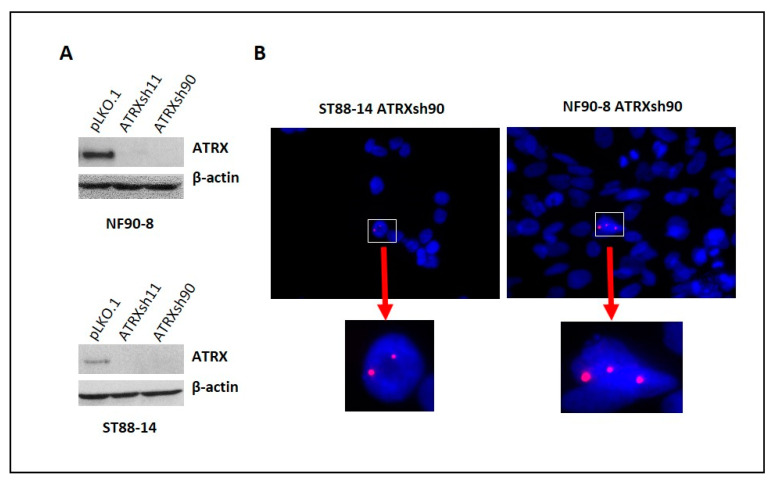
ATRX knockdown resulted in rare abnormal telomere FISH signals: Reduction in ATRX expression using ATRX shRNAs in NF90-8 and ST88-14 MPNST cell lines (**A**), empty lentiviral vector (pLKO.1) was used as control. Development of ultrabright telomere foci reminiscent of ALT was detected with telomere FISH after ATRX knockdown (**B**).

**Figure 5 cancers-14-03015-f005:**
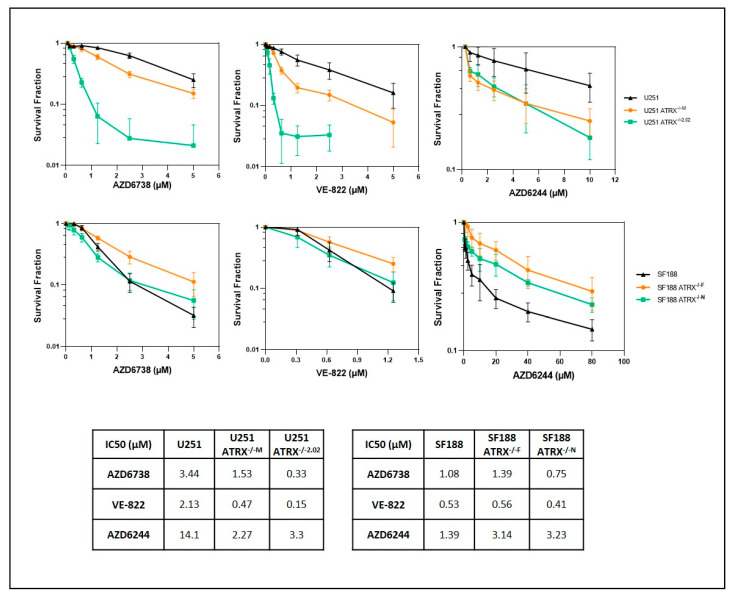
ALT-positive, ATRX-deficient U251 cells display an increased sensitivity to ATR inhibition. U251 and SF188 cells were treated with various doses of AZD6244, VE-822, or AZD6738 for 5 days, cell survival was normalized with vehicle control. ATR inhibitors (AZD6738 and VE-822) decreased growth of U251 (ALT+) (top row), but not in cell line SF188 (ALT-) (bottom row). MEK inhibition (AZD6244, selumetinib) had a modest effect on growth in U251 as well. IC50 of drugs are listed at the bottom. Data are presented as mean ± s.d.

**Figure 6 cancers-14-03015-f006:**
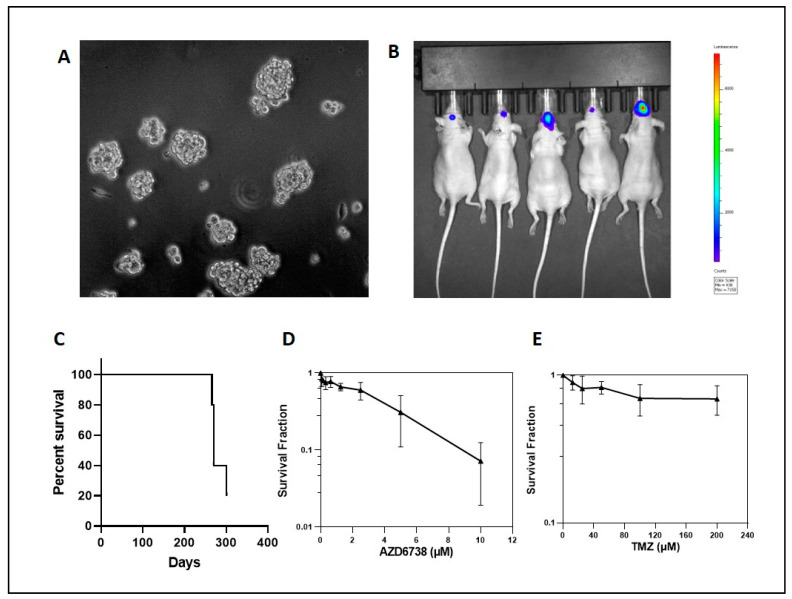
ATR inhibition impairs growth of the NF1-patient-derived glioblastoma cell line JHH-NF1-GBM1 in vitro: JHH-NF1-GBM1 grown under neurosphere culture conditions (**A**) and in orthotopic nude mouse xenografts (**B**,**C**). Decreased cell growth after treatment with the ATR inhibitor, AZD6738, was noted in vitro (**D**); although, these cells remained relatively resistant to treatment with temozolomide (**E**).

**Figure 7 cancers-14-03015-f007:**
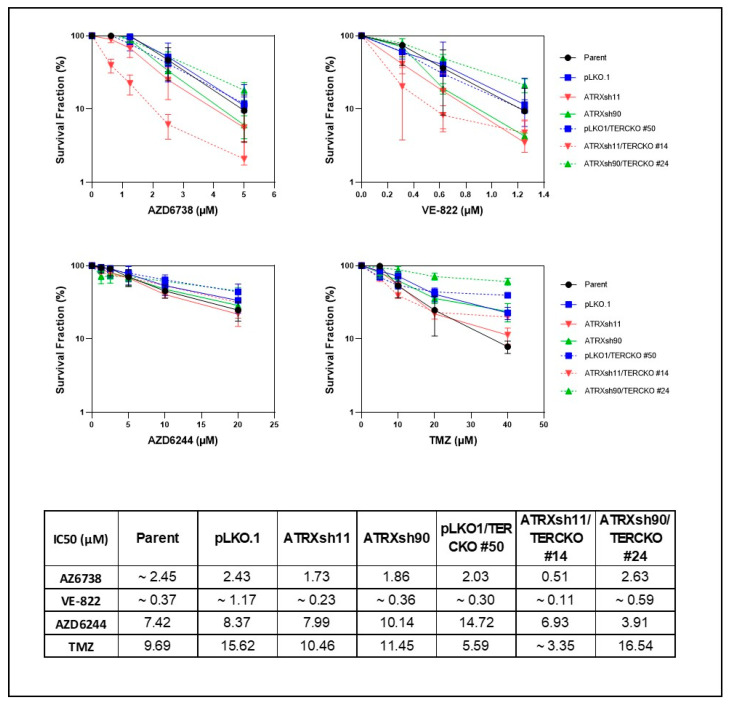
ATRX and TERC loss impairs growth of the MPNST cell line NF90-8 when treated with ATR inhibitors. MPNST cell clones with concurrent ATRX knockdown and TERC knockout are sensitive to ATR inhibitors AZD6738 and VE-822, but not to MEK inhibitor (AZD6244) or temozolomide. IC50 of drugs are listed at the bottom.

**Figure 8 cancers-14-03015-f008:**
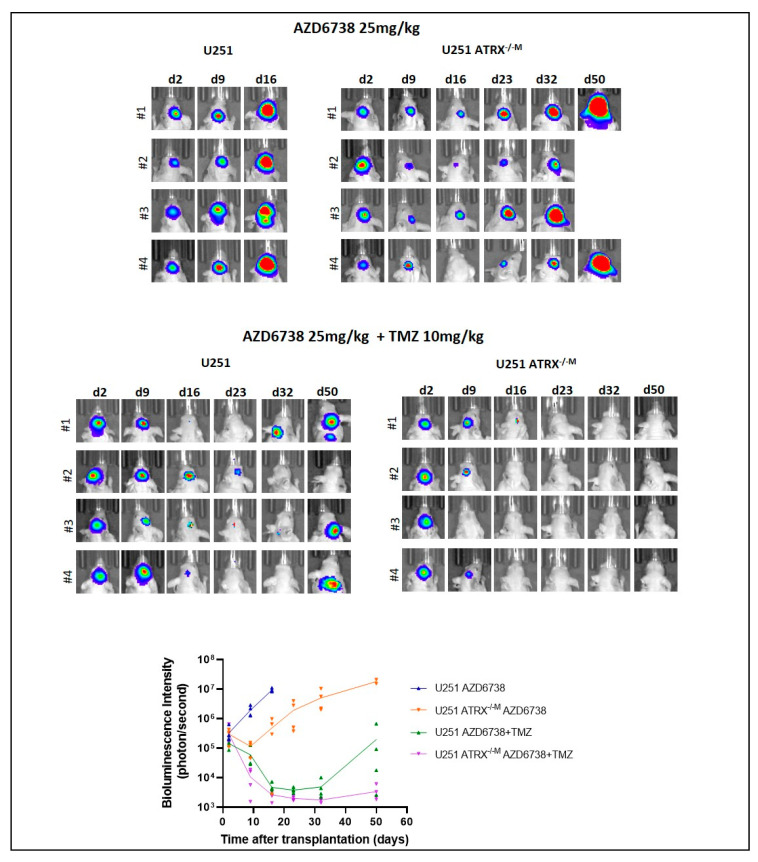
ATR inhibition transiently impairs growth in the NF1 glioma cell line U251 with ATRX loss. Orthotopic xenografts using the glioma cell line U251 demonstrate transient growth inhibition after treatment with the ATR inhibitor, AZD6738. Growth inhibition is more persistent when combined with temozolomide.

## Data Availability

Not applicable.

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
