# Peer review of "Therapeutic Vulnerability to ATR Inhibition in Concurrent NF1 and ATRX-Deficient/ALT-Positive High-Grade Solid Tumors"

_cancers, 2022, doi:10.3390/cancers14123015_

Round 1
Reviewer 1 Report
Authors enough responded to comments by reviewer. Current version could be accepted for publication.
Reviewer 2 Report
The authors have incorporated all comments.
Reviewer 3 Report
The authors have adequately addressed my concerns. This article is now acceptable for publication.
This manuscript is a resubmission of an earlier submission. The following is a list of the peer review reports and author responses from that submission.
Round 1
Reviewer 1 Report
In the current manuscript, it is very interesting to analyze the function of ATRX in Neurofibromatosis type I, and their experimental results suggest the importance of ATRX for the cell proliferation of NF1-related cells. However, their analysis did not provide any information for understanding the molecular mechanism how ATRX deficiency causes the apoptotic cell death.
(1) In Figure 1 and 4, to evaluate the situation of telomere, telomere-specific FISH analysis was utilized. However, this method has a problem in terms of quantification. It would be better to utilize another quantitative method.
(2) In Figure 2 and 3, ATRX knockdown in the human glioma cell line caused the increase of apoptotic cells. How about the molecular mechanism for the apoptotic cell death.
(3) Figure 3 should be re-desinated. (A) to (C) could not be found in the figure.
(4) In Figure 5 and 7, graphs, especially the measure in graphs should be re-designated, and log-scale should be much better. It is recommended to calculate ED50 of each compound utilized in current results.
(5) In Supplementary Figure 8, the expression of p21 was analyzed, however other p53 target genes were not. How about the expression of them, for example, how about the expression of Puma and other Bcl family proteins.
(6) In the legend for Figure 6, the explanation for Figure 6C was not found.
(7) In Figure 8, all data should be shown in the graph.
Reviewer 2 Report
The authors present an understudied pathway in NF1 malignancies and show potential for its implementation using both in vitro and in vivo evidence, especially in glioma cell lines.
The authors could elaborate further in their discussion why they think the role could be in MPNST/sarcoma's related to NF1 as aberrant ATRX as been shown to be a poor prognosticator in NF1 associated MPNSTs (PMID: 29796169)
Reviewer 3 Report
Although the topic is interesting in this field, the manuscript is overall hard to follow, and the hypothesis is not clear. Specific commons are shown below.
- The title did not represent the key finding of this study as ALT presence is more associated with the sensitivity of ATR inhibitor in concurrent NF1- and ATRX-deficient high-grade solid tumors.
- Since the authors used different models in this study, it would better to use one or two sentences to explain why specific glioma cells were chosen for specific experiments for better transition in each session.
- Figure 1 and 4: Please provide the quantification data of increased telomere FISH signal or ALT-like foci.
- Figure 4 and 5: What is the difference between U251 and SF188 high-grade glioma cell lines besides ALT? Please also provide the basal ALT-like foci of each cell line used in this study if the ALT is the determine factor for the sensitivity of ATR inhibition in these NF1- and ATRX-deficient high-grade glioma cell lines.
- Figure 5: It is also not clear why the authors wanted to test MEK inhibition (AZD6244) in these cell lines.
- Figure 8: Please provide the signal intensity data to show the tumor growth. ATR inhibitor sensitized U251 cells to temozolomide in vivo. How about other models, e.g., JHH-NF1-GBM1 or NF90-8 models?
- NF1 and ATRX deficiency induce high replication stress, which can increase the sensitivity of ATR/CHK1 pathway inhibition. Did the authors observe any increased replication stress (i.e. increased gammaH2AX and RPA foci) in NF1- and ATRX-deficient glioma cells? Are these NF1- and ATRX-deficient glioma cells also sensitive to CHK1 inhibitors besides ATR inhibitors?